# Cross-cultural adaptation and psychometric evaluation of the Yoruba version of Oswestry disability index

Chidozie Emmanuel Mbada[1,2], Oluwabunmi Esther Oguntoyinbo[1], Francis Oluwafunso Fasuyi[3], Opeyemi Ayodiipo Idowu[4]*, Adesola Christiana Odole[2], Olusola Ayanniyi[2], Olubusola Esther Johnson[1], Elkanah Ayodele Orimolade[5], Ajibola Babatunde Oladiran[6], Francis Fatoye[7]

1 Department of Medical Rehabilitation, College of Health Sciences, Obafemi Awolowo University, Ile–Ife, Nigeria, 2 Department of Physiotherapy, Faculty of Clinical Sciences, College of Medicine, University of Ibadan, Ibadan, Nigeria, 3 Department of Physiotherapy, Faculty of Allied Health Sciences, University of Medical Sciences, Ondo, Nigeria, 4 Department of Physiotherapy, School of Basic Medical Sciences, College of Medical Sciences, University of Benin, Benin-City, Nigeria, 5 Department of Orthopaedic Surgery and Traumatology, College of Health Sciences, Obafemi Awolowo University, Ile–Ife, Nigeria, 6 Department of Orthopaedic Surgery and Traumatology, College of Medicine, University of Ibadan, Ibadan, Nigeria, 7 Department of Health Professions, Faculty of Health, Psychology and Social Care, Manchester Metropolitan University, Manchester, United Kingdom

* opeyemi.idowu@uniben.edu

**Data Availability Statement:** All relevant data are within the manuscript and its Supporting Information files.

## Abstract

### Introduction

Low-Back Pain (LBP) is a common public health problem that is often worsened by mal-adaptive beliefs and disability. Thus, necessitating the need for availability of outcome measures to assess these sequelae among patients with chronic LBP. This study aimed to cross-culturally adapt and determine the psychometric properties of the Yoruba version of the ODI (ODI-Y).

### Methods

The ODI-Y was cross-culturally adapted following the process involving forward translation, synthesis, backward translation, expert review, and pilot testing. Internal consistency and test-retest reliability of the ODI-Y were determined using the Cronbach's alpha and intra-class correlation. Other psychometric properties explored included the factor structure, convergent validity, standard error of measurement and the minimal detectable change.

### Results

One hundred and thirty-six patients with chronic LBP took part in the validation of the ODI-Y; 86 of these individuals took part in the test-retest reliability (within 1-week interval) of the translated instrument. The mean age of the respondents was 50.5±10.6years. The ODI-Y showed a high internal consistency, with a Cronbach's alpha (α) of 0.81. Test-retest of the Yoruba version of the ODI within 1-week interval yielded an Intra-Class Correlation coefficient of 0.89. The ODI-Y yielded a three-factor structure which accounted for 61.56% of the

**Funding:** The authors received no specific funding for this work.

**Competing interests:** The authors have declared that no competing interests exist.

variance. Correlation of ODI-Y with the visual analogue scale was moderate (r = 0.30; p = 0.001). The standard error of measurement and minimal detectable change of the ODI-Y were 2.0 and 5.5.

## Conclusions

The ODI was adapted into the Yoruba language and proved to have good psychometric properties that replicated the results of other obtainable versions. We recommend it for use among Yoruba speaking patients with LBP.

## Introduction

Low-Back Pain (LBP) is a major public health challenge with a high disability burden [1]. According to the 2010 Global Burden of Disease Study, LBP is among the top 10 diseases and injuries that account for the highest number of disability-adjusted life-years worldwide [2]. As a result of this, outcome tools that assess the disability resulting from LBP have become more abundant. Among the outcome tools, Roland-Morris Disability Questionnaire and Oswestry Disability Index (ODI) [3–5] are mostly recommended [6] owing to abundant reports literature on their clinimetric and psychometric properties [3,5,7,8].

Researchers and clinicians often use the Oswestry Disability Index (ODI) as a disease-specific questionnaire to assess pain and disability resulting from LBP [3,5,7]. Fairbank et al. developed the ODI as a self-administered 10-item questionnaire [8]. The ODI assesses the consequences of pain on typical daily activities, including personal care, lifting, walking, sitting, standing, sleeping, sex life, social life, and travelling [8]. The anchors of the tool vary from no disability (0) to maximum disability (100) [8]. Based on psychometric properties and clinical usability, various languages translations including the Greek [9], Norwegian [10], Japanese [11], Turkish [12], Korean [13], Arabic [14], German [15], Danish [16], Iranian [17], Brazilian-Portuguese [18], Italian [19] and Tamil [20] exist. Most of these translations report excellent psychometric properties. The ODI has a Cronbach $\alpha$ ranging from 0.71 to 0.87, an intraclass correlation coefficient from 0.84 to 0.94 and a test-retest reliability value between 0.83 and 0.99 [7].

As cultural groups vary in disease perception and expressions and their use of various health care systems, local languages enhance the comprehensibility of outcome tools. [21]. Thus, increasing the comprehensibility and usability of outcome tools, especially among non-English speakers necessitated the translation of outcome tools or questionnaires into local languages. Further, patients find outcomes translated into their local languages as easily accessible, user-friendly, and comprehensible [22]. To date, only one translation (Hausa version) of the ODI with requisite data on validity and reliability exist [23], thus the need for translations of the ODI in other Nigeria languages.

Although English is the official language in Nigeria, a sizeable number of Nigerian patients are not literate in English [22]. Nigeria, as the most populous black African nation, is a multiethnic and multi-lingual country but with three major ethnic groups (Hausa, Igbo, and Yoruba) and with different languages. The Yoruba tribe makes up close to 40 million people [24], this should be among the largest ethnic groups of sub-Saharan Africa. Besides, other countries including the Benin Republic, Togo and Brazil speak the Yoruba language. [25,26]. Therefore, the availability of ODI in the Yoruba language will improve the uptake of the tool among Yoruba speaking patients with LBP. This study aimed to cross-culturally adapt, test the convergent

validity, small detectable change, factor structure, ceiling and floor effects and test-retest reliability of the ODI among patients with LBP.

## Materials and methods

### Ethical approval and informed consent

The Health Research and Ethics Committee of the Obafemi Awolowo University Teaching Hospitals Complex, Ile-Ife, Nigeria gave ethical approval for this study (ERC/2015/12/05) (for a period between 10/12/2015 and 10/01/2016). The respondents also gave their written informed consents prior to participation in the study. Further, the respective heads of departments of the selected hospitals gave administrative permission to conduct the study.

### Study design

Cultural adaptation, test-retest and cross-sectional psychometric analyses.

### Instruments

**The Oswestry disability questionnaire.** The ODI questionnaire is a ten 6-point questionnaire. The first segment of the tool assesses the intensity of pain, while the remaining sections assess the disabling effect of pain on typical daily activities such as personal care, lifting, walking, sitting, standing, sleeping, sex life, social life, and traveling. Each item has scores ranging from 0 to 5, with the sum of scores of the 10 items expressed as a percentage of the maximum scores, varying from 0 (no disability) to 100 (maximum disability). Typically, it takes about five minutes to complete the questionnaire and less than one minute to compute scores [3].

**The Yoruba version of the Visual Analogue Scale (VAS).** The VAS represents the intensity dimension of pain by a 10 cm line with two anchors of "no pain" and "worst pain I ever felt" [27]. The VAS assesses pain intensity, has excellent psychometric properties, and has wide applicability in clinical and research settings [27–30].

Odole and Akinpelu [29] reported a moderate correlation between the English version and the translated Yoruba version of the VAS.

### Cultural adaptation of the ODI to the Yoruba Language

Using a five-step guideline proposed by Guillemin, Bombardier, and Beaton [21], the English version of the ODI questionnaire was translated into the Yoruba language. The translation process in sequential order comprised:

i.  Forward translation of the items and response choices of the English version of the ODI to the Yoruba language by two professionally qualified translators who are both native speakers of Yoruba language and bilingual in Yoruba and English languages. One translator had information about the concepts being examined in the questionnaire. This stage involved two forward translations referred to as T1 and T2.

ii.  Synthesis: Synthesis: The two translators and the researcher (CEM) produced a synthesized version (T3) following a reconciliation meeting.

iii.  Back translation: Back translation: Two independent qualified English translators translated the synthesized version (T3) back into the English language (BT1 and BT2). They individually identified inconsistencies in the words and concepts of the synthesized version.

iv. Expert committee review: An expert committee comprising three of the researchers (CEM, OEO, and, OEJ, physiotherapists by profession) and all four translators met to discuss issues of cultural adaptations and linguistic equivalence with the original English version of ODI. The meeting produced the final version of the Yoruba ODI (T4). The expert committee made some adaptations to the ODI while translating it from the original English version. Some adaptations were made to the ODI-Y while translating it from the original English version. Specifically, in section one (items 2 and 3), the Yoruba word *àfaradà* was used instead '*dédé*' (which means moderate) which should have been the most suitable transliteration equivalent. However, using '*dédé*' in the context will not make a meaningful sentence. In section three (Abala kéta), the word '*Gbígbé*' had to be qualified in the ODI-Y, with '*Nnkan*' to become '*Nnkan gbígbé*' which means lifting. Also, item 5 in section three, was translated in the passive form, as a direct translation in the active cast may convey a different meaning, apart from that intended in the original translation. In section five (Abala Karùn-ún), item 1, the term 'favourite chair' was changed to 'comfortable chair' because the term favourite chair is not commonly used in this study context. In section seven (Abala Kéje), items 2, 3 and 4, the element of time translated as *àsìkò* was included to trade the sense of sleep duration missing in the literal equivalent of the translation in the Yoruba language.

v. Pilot testing: Fifteen Yoruba speaking patients with LBP filled the pre-final version of the ODI (T4). The patients also undertook individualized cognitive debriefing. The cognitive debriefing was to explore the respondents' perception, understanding, interpretation of various terminologies used, and the formatting of the translated items of the T4. Analysis of the participants' interpretation of items evaluated whether or not the adapted version retained equivalence of the items in the English version. Reports were prepared at each stage to cover issues that were faced and how they were resolved.

## Psychometric testing

There is no consensus about the minimum required sample size for validation studies. However, no less 50 participants is considered adequate for construct validity, reliability, and ceiling/floor effects analyses [31]. Based on sample size ranges in previous studies on translation of the ODI, a sample range of between 30 and 126 [19, 20] was observed. Thus, a sample size estimate of 150 participants was considered adequate in this study. All the respondents in this study were recruited from three hospitals in the South-west zone of Nigeria namely: Obafemi Awolowo University Teaching Hospital complex Ile-Ife (OAUTHC), Wesley Guild Hospital, Ilesha, and University College Hospital, Ibadan. Eligibility for inclusion in the study was having a history of non-specific LBP of three months and longer, being literate in Yoruba languages, and having no cognitive impairment. The diagnostic criteria for non-specific LBP included the absence of serious pathology (red flags conditions such as fracture, malignancies or infection) and radicular syndrome. Volunteers with non-specific chronic LBP but with a systemic illness, rheumatologic diseases or other co-morbidity were excluded from the study. The ODI-Y and the VAS were administered on the participants on the same day. In addition to this, socio-demographic information and anthropometric measurements were also taken. Out of the 150 consenting patients with chronic LBP consulted for the cross-sectional study, only 136 (70 males and 66 females) returned their ODI-Y questionnaires validly completed. Eighty-six respondents (55.9±11.0 years; 54.7% female) completed the ODI-Y again after seven days of the first administration. The test-retest sample was a subset of the sample that took part in the validity assessment of the ODI and also consented to partake in the test-retest phase of the study.

## Data analysis

Data were assessed for normality using visual (normal distribution curve and Q-Q plot) and statistical methods (Shapiro-Wilk's test and Skeweness/Kurtosis scores). Data were summarized using descriptive statistics of mean, standard deviation, percentages and median.

Validity: Construct validity describes the extent to which an outcome is able to measure a construct it was intended to measure. To our knowledge, there is no "gold standard" measure of the ODI and therefore it is not possible to assess criterion validity. However, construct validity of the ODI-Y was determined by correlating the ODI-Y scores with the Yoruba language VAS. The construct validity was assessed using the Spearman's correlation coefficients (data was not normally distributed), and was rated as weak (0–0.2), moderate (0.3–0.6), and strong (0.7–1.0) [32]. The choice of the VAS in the validity assessment of the ODI was informed by the established relationship between the two in the literature [33, 34].

Exploratory factor analysis (EFA), which shows the number of items of a scale that go together, was used to determine the factor structure of the ODI-Y [35]. Kaiser-Meyer- Olkin value, Bartlett's test of sphericity and correlation matrix table was used to check the suitability of the ODI-Y data prior to the conduction of principal component analysis (PCA). The authors set the minimum eigen value for factor retention to be $\geq 1.0$. Retained and excluded factors were also explored using parallel analysis. Promax (oblique) rotation, which assumes that factors can be related, was done with factor loadings less than 0.4 suppressed. Extraction was done using principal axis factoring. The number of factors and the interrelationships between the items loading in each factor were then compared with the factor structures of the previous ODI to find out whether there were differences owing to dissimilar population characteristics.

The reliability of the ODI-Y (an indication of how the instrument measures consistently over time) was determined using the Intra-Class Correlation (ICC). The absolute agreement, 2-way random-effects approach which assumes that errors in measurement could arise from either raters or participants) was used for the test-retest reliability of the ODI-Y. An ICC in the range of 0.4–0.75 was regarded as moderate, while values below and above this range were considered low and high respectively [36]. Reliability was also evaluated using the standard error of measurement (SEM) and minimal detectable change (MDC). Minimal detectable change is defined as the amount of change in a score that is required to distinguish a true performance change from a change due to chance [37]. The MDC was calculated using the standard error of measurement (which is based on the standard deviation of observed test scores for a given true test score). The standard error of measurement of the ODI-Y was calculated using the formula: $SEM = SD\sqrt{1-R}$ [33]. Further, the MDC of the ODI-Y was calculated with the formula: $MDC = 1.96 \times \sqrt{2} \times SEM$ [37]. Bland-Altman analysis [38] was also used to visually assess heterodascity between test-retest measurements by plotting mean scores against difference in total scores. Cronbach alpha was used to test for the internal consistency of the ODI-Y respectively. A Cronbach's alpha not less 0.7 is recommended for outcome measures [39]. Potential ceiling and floor effects were considered present if >15% of respondents achieved the lowest (10%) or highest possible total scores (100%) [31]. Data were analysed using SPSS (Statistical Package for Social Sciences) for Windows (Version 16.0. Chicago, SPSS Inc.) Alpha level was set as 0.05.

## Results

The age of the 131 participants ranged from 35–70 years. The mean age, weight, height and BMI of the respondents (51.5% females) was 50.7±10.6years, 75.0±11.2Kg, 1.67±0.04m, and 26.71±4.23Kg/m$^2$ respectively. The general characteristics of the respondents by gender are

**Table 1. General characteristics of the participants by gender (N = 136).**

| Variables | Male Mean ± SD | Female Mean ± SD | t-cal | p-value |
|---|---|---|---|---|
| Age (years) | 48.5 ± 10.7 | 52.7 ± 10.2 | -2.328 | 0.021 |
| Weight (kg) | 74.9 ± 10.9 | 75.1 ± 11.5 | -0.083 | 0.834 |
| Height (m) | 1.68 ± 0.04 | 1.68 ± 0.04 | 0.706 | 0.482 |
| BMI (Kg/m$^2$) | 26.6 ± 4.21 | 26.8 ± 4.28 | -0.255 | 0.799 |

SD: Standard deviation; BMI: body mass index

presented in Table 1. Shapiro-Wilk's normality test (p < 0.05), as well as, the Q-Q plots observation, showed that the ODI-Y was not normally distributed.

The Spearman's rank correlation coefficient for the convergent validity of the ODI-Y with the VAS was r = 0.30; p = 0.001. The 1-week test-retest reliability of the ODI-Y using ICC was 0.80 (95% CI 0.74–0.84). Further, the internal consistency of the ODI-Y was 0.81. The Item by Item Correlation between the Test-Retest of the ODI-Y and the Cronbach's Alpha if an item of the ODI-Y is deleted are presented in Tables 2 and 3 respectively. The SEM and MDC of the ODI-Y were 2.0 and 5.5. The mean difference between the test and retest scores as shown by Bland-Altman analysis was -0.26. Further, only 2 outliers affected the 95% limits of agreements.

Principal component analysis (PCA) with Promax rotation was used to evaluate the factor structure of the ODI-Y. To determine that the data was suitable for factor analysis, indicators including the correlation matrix table (presence of many coefficients > 0.3), Kaiser-Meyer-Olkin measure of sampling adequacy (0.74) and Bartlett's test of sphericity ($X^2$ = 432.34, p < 0.001) were considered; all of them indicated that PCA could proceed. Only factors with eigen value >1 were considered to contribute significantly to explaining variance. Factors loading > 0.3 were included in the model. Principal components extraction yielded a total of three factors which accounted for 61.56% of the total variance of the 10 factors. The first factor, with an eigenvalue of 3.9, consisted of items 2, 4, 6, 8, 9 and 10 accounting for 39.5% of the variance. The second factor, with an eigenvalue of 1.2, consisted of items 5 and 7 accounting for 12% of the variance. The third factor with an eigenvalue of 1.0 consisted of items 1 and 3 accounting for 10.1% of the variance. This is presented in Table 4.

**Table 2. Reliability of the Yoruba version of the ODI.**

| Global score of the ODI (α) | 0.81 |
|---|---|
| Item | Cronbach's alpha if Item Deleted |
| 1 | 0.814 |
| 2 | 0.783 |
| 3 | 0.806 |
| 4 | 0.781 |
| 5 | 0.798 |
| 6 | 0.775 |
| 7 | 0.80 |
| 8 | 0.775 |
| 9 | 0.775 |
| 10 | 0.784 |

ODI: Oswestry disability index; α; Cronbach's alpha.

**Table 3. Test-retest of the Yoruba version of the ODI.**

|  | ICC | 95% CI |
|---|---|---|
| Global score | 0.80 | 0.74–0.84 |
| Item by item |  |  |
| 1 | 0.876 | 0.80–0.92 |
| 2 | 0.917 | 0.872–0.946 |
| 3 | 0.971 | 0.955–0.981 |
| 4 | 0.939 | 0.906–0.96 |
| 5 | 0.969 | 0.952–0.98 |
| 6 | 0.94 | 0.911–0.962 |
| 7 | 0.893 | 0.833–0.931 |
| 8 | 0.929 | 0.891–0.954 |
| 9 | 0.900 | 0.846–0.935 |
| 10 | 0.945 | 0.915–0.964 |

ODI: Oswestry disability index; ICC: intra-class correlation; CI: confidence interval

The ODI-Y had no ceiling or floor effect as no respondent had the maximum possible score and only 2.2% of respondents had the minimum possible score.

## Discussion

To the best of our knowledge, this is the first study that cross-culturally adapted the ODI-Y and determined its psychometric properties. The respondents in this study were patients with chronic LBP with whom maladaptive illness beliefs and disability are common [40, 41]. Disability associated with chronic LBP is an intricate and multidimensional phenomenon [42] that further perpetuates chronicity of LBP [43, 44]. Thus, disability resulting from LBP is associated with difficulty in performing almost all basic and instrumental activities of daily living [40, 45].

To attain comprehensive multidimensional assessment of the burden, as well as, intervention outcomes, tools such as the RMDQ and ODI have enjoyed widespread application in

**Table 4. Principal component analysis of the Yoruba version of the ODI.**

| Item | Factor 1 | Factor 2 | Factor 3 | Communality |
|---|---|---|---|---|
| 1. Pain | 0.118 | -0.281 | 0.813[a] | 0.703 |
| 2. Personal care | 0.657[a] | 0.017 | 0.088 | 0.484 |
| 3. Lifting | -0.095 | 0.383 | 0.741[a] | 0.735 |
| 4. Walking | 0.643[a] | 0.150 | 0.006 | 0.516 |
| 5. Sitting | 0.113 | 0.608[a] | 0.090 | 0.472 |
| 6. Standing | 0.793[a] | -0.014 | 0.026 | 0.633 |
| 7. Sleeping | 0.021 | 0.883[a] | -0.141 | 0.769 |
| 8. Sex | 0.715[a] | -0.074 | 0.200 | 0.593 |
| 9. Social life | 0.668[a] | 0.065 | 0.126 | 0.554 |
| 10. Travelling | 0.864[a] | 0.023 | -0.302 | 0.698 |
| Eigenvalue | 3.95 | 1.20 | 1.0 |  |
| % of the variance explained | 39.47 | 12.0 | 10.0 |  |

[a]Principal component coefficient $\geq$ 0.4

ODI: Oswestry disability index

clinical and research settings [46, 47], owing to the good to excellent psychometric properties of the original versions, variants and translations of these tools [48, 49].

The finding of the current study on the convergent validity of ODI-Y with VAS shows a weak but significant correlation. This finding is consistent with previous results on ODI translations, showing significant correlation between translated versions of ODI and pain tools with correlation co-efficient values ranging weak to high. For example, the correlation co-efficient for Norwegian (0.52) [10], Korean (0.42) [13], Swiss-German (0.78) [15], Iranian (0.54) [17] and Brazilian-Portuguese (0.66) [18] versions of the ODI, showed weak to high correlation with pain intensity. The positive but weak correlation between ODI-Y and pain intensity observed in this study may suggest that the new tool is valid for measuring the presence functional disability rather than severity of pain.

The test-retest of the ODI-Y within 1-week interval showed a high correlation based on the ICC value. High ICC coefficient obtained in this study conforms with the recommendation of an ICC of 0.75 or more, considered in many studies as reliable [8]. The narrow 95% CI obtained for the ICCs in this study confirms that the ODI-Y can yield reliable results when administered on multiple occasions. Thus, the results on test-retest reliability obtained in this study are comparable with those reported in the Norwegian [10], Korean [13], and Brazilian-Portuguese [18] versions, where an ICC range of between 0.7 and 0.99 has been observed for intervals ranging from 2 days to 4 weeks [50].

Another finding of this study indicate that SEM and MDC of the ODI-Y were 2.0 and 5.5, while the by Bland-Altman analysis showed a mean difference of -0.26 between the test and retest scores. The estimated SEM (2.0) of the ODI-Y resulted in a $MCD_{95\%}$ of 5.5. The $MDC_{95\%}$ found in our study was similar to that reported in the Croatian version of the ODI (6.0) [50]. The MDC of the ODI-Y was lower than that of the Polish (MDC = 10) [51]; German (9.0) [15]; Hungarian (MDC = 11) [52] and Chinese (12.8) [53] translations of the ODI. An MDC of 5.5 found in our study implies that below 5.5, the measurement error of the ODI-Y is indistinguishable. That only 2 outliers affected the 95% limits of agreements during the Bland-Altman analysis indicates a very strong agreement between the test and retest scores and minimal within-subject variations. The Bland-Altman analysis accounts for the shortcoming of the ICC which might indicate strong correlations between two measurements with minimal agreement [54]. The findings of this study show that the ODI-Y had a high internal consistency. A higher internal consistency > 0.95 would have indicated a redundancy in the questionnaire items. The internal consistencies of most of the ODI translations [49, 50, 55] fall within this band.

Principal component analysis of the ODI-Y revealed a three-factor structure accounting for 61.56% of the variance. The first factor (everyday activities) includes personal care, walking, sitting, standing, sex life, social life, and travelling, second factor (sedentary activities) includes sleeping and sitting while the third factor (pain/lifting) includes pain and lifting. Most factor analyses of the language translations of the ODI yielded one factor [5, 15, 56, 57, 58] or two-factor structures [50, 59, 60, 61]. Such factors reported in previous two-factor models include social/ recreational activities and non-recreational activities [50], dynamic and static activities [59, 62], pain-related activity, and pain intensity and pain-related participation [55]. Gabel and colleagues [62] conducted a PCA of the ODI in a large sample of patients with LBP derived from the international Spine Tango registry of EUROSPINE. Their analysis yielded a single-factor model which was confirmed by the CFA. They further conducted a CFA on the literature-recommended two-factor model of the ODI; this yielded indicators which were not within the ranges of acceptable fit. Based on the evidence from the study by Gabel et al [62], that none of the previous two-factor models have similar factor loading, and the results from our study, it is recommended that a global score of the ODI be used in research and the clinical

settings. The unidimensionality of the ODI, however, remains debatable. Larger sample size studies are thus needed to provide answers to the dimensionality of the ODI. Furthermore, the findings of this study indicate that the ODI-Y had no floor or ceiling effects. Floor and ceiling effect refers to the percentage of patients scoring maximal or minimal scores. It is recommended that questionnaires with more than 15% of the respondents scoring either the maximal or minimal scores should not be used. Thus, application of ODI-Y seems not to be limited by any concern of floor or ceiling effects. In sum, the ODI-Y showed acceptable internal consistency, test-retest reliability, convergent validity, a three-factor structure, and no floor or ceiling effects. This study is not without potential limitations. Firstly, generalizability of the findings to other population other than those with chronic LBP should be treated with caution. Further, a not large enough sample size renders a Rasch analysis of the ODI-Y not meaningful, thus losing vital results that bother on unidimensionality, appropriateness of response categories for each item (category function) and equivalence of items on meaning to different respondents (differential item functioning) that may have been gotten from the analysis. Overall, the ODI-Y is recommended for assessing patients with chronic LBP among the Yoruba population.

## Conclusion

The Yoruba version of the ODI questionnaire is valid and reliable, with adequate psychometric properties, and it can be used in Yoruba speaking patients with chronic LBP. The psychometric properties of the ODI-Y are comparable with the original English and other translations of the ODI.

## Supporting information

**S1 Data. The ODI validity data (n = 136).**
(XLSX)

**S2 Data. The ODI-Yoruba test-retest reliability data.**
(XLSX)

**S1 Text. Ìgbéléwòn Bèbèré Èyìn Dídùn ti Oswestry (The Yoruba Oswestry Disability Index).**
(DOCX)

## Acknowledgments

The authors thank all the people who volunteered to participate in the study.

## Author Contributions

**Conceptualization:** Chidozie Emmanuel Mbada, Oluwabunmi Esther Oguntoyinbo, Francis Fatoye.

**Data curation:** Francis Oluwafunso Fasuyi, Opeyemi Ayodiipo Idowu.

**Formal analysis:** Francis Oluwafunso Fasuyi, Opeyemi Ayodiipo Idowu.

**Investigation:** Chidozie Emmanuel Mbada, Oluwabunmi Esther Oguntoyinbo, Francis Oluwafunso Fasuyi, Adesola Christiana Odole, Olubusola Esther Johnson, Ajibola Babatunde Oladiran.

**Methodology:** Chidozie Emmanuel Mbada, Opeyemi Ayodiipo Idowu, Adesola Christiana Odole, Olusola Ayanniyi, Elkanah Ayodele Orimolade, Francis Fatoye.

**Project administration:** Chidozie Emmanuel Mbada, Olusola Ayanniyi, Olubusola Esther Johnson, Francis Fatoye.

**Resources:** Chidozie Emmanuel Mbada, Adesola Christiana Odole, Elkanah Ayodele Orimolade, Ajibola Babatunde Oladiran, Francis Fatoye.

**Software:** Francis Oluwafunso Fasuyi, Opeyemi Ayodiipo Idowu, Ajibola Babatunde Oladiran.

**Supervision:** Chidozie Emmanuel Mbada, Opeyemi Ayodiipo Idowu, Olusola Ayanniyi, Olubusola Esther Johnson, Ajibola Babatunde Oladiran, Francis Fatoye.

**Validation:** Chidozie Emmanuel Mbada, Oluwabunmi Esther Oguntoyinbo, Adesola Christiana Odole, Olusola Ayanniyi, Olubusola Esther Johnson, Elkanah Ayodele Orimolade, Ajibola Babatunde Oladiran.

**Visualization:** Chidozie Emmanuel Mbada, Elkanah Ayodele Orimolade, Ajibola Babatunde Oladiran, Francis Fatoye.

**Writing – original draft:** Chidozie Emmanuel Mbada, Oluwabunmi Esther Oguntoyinbo, Francis Oluwafunso Fasuyi, Olubusola Esther Johnson, Elkanah Ayodele Orimolade, Ajibola Babatunde Oladiran.

**Writing – review & editing:** Chidozie Emmanuel Mbada, Oluwabunmi Esther Oguntoyinbo, Francis Oluwafunso Fasuyi, Opeyemi Ayodiipo Idowu, Adesola Christiana Odole, Olusola Ayanniyi, Olubusola Esther Johnson, Elkanah Ayodele Orimolade, Ajibola Babatunde Oladiran, Francis Fatoye.

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
