## [Decision Letter · Decision Letter 0]

5 Nov 2019

PONE-D-19-21244

Cross-cultural adaptation and psychometric evaluation of the Yoruba version of Oswestry disability index

PLOS ONE

Dear Dr. Idowu,

Thank you for submitting your manuscript to PLOS ONE. After careful consideration, we feel that it has merit but does not fully meet PLOS ONE’s publication criteria as it currently stands. Therefore, we invite you to submit a revised version of the manuscript that addresses the points raised during the review process.

We would appreciate receiving your revised manuscript by Dec 20 2019 11:59PM. To enhance the reproducibility of your results, we recommend that if applicable you deposit your laboratory protocols in protocols.io, where a protocol can be assigned its own identifier (DOI) such that it can be cited independently in the future. For instructions see: http://journals.plos.org/plosone/s/submission-guidelines#loc-laboratory-protocols

We look forward to receiving your revised manuscript.

Kind regards,

Ali Montazeri

Academic Editor

PLOS ONE

Journal Requirements:

3. Please carefully proofread your manuscript for typographical errors. For example, on page 8 “However, no less 50 participants be considered adequate…” should be “However, no less than 50 participants is considered adequate…”

4. Please state in your methods section the participant recruitment date.

5. Please state the participants age range (in addition to their mean age).

6. Please amend your current ethics statement and methods section to address the following concerns:   a) Did participants provide their written or verbal informed consent to participate in this study?  b) If consent was verbal, please explain:  i) Why was written consent not obtained?   ii) How did you record/document participant consent?  iii) Did the ethics committees/IRBs approve this consent procedure?

7. Please amend the manuscript submission data (via Edit Submission) to include author Ajibola Babatunde Oladiran.

Reviewers' comments:

Reviewer's Responses to Questions

**Comments to the Author**

1. Is the manuscript technically sound, and do the data support the conclusions?

Reviewer #1: Partly

Reviewer #2: Partly

2. Has the statistical analysis been performed appropriately and rigorously? 

Reviewer #1: Yes

Reviewer #2: No

3. Have the authors made all data underlying the findings in their manuscript fully available?

Reviewer #1: Yes

Reviewer #2: Yes

4. Is the manuscript presented in an intelligible fashion and written in standard English?

Reviewer #1: Yes

Reviewer #2: Yes

5. Review Comments to the Author

Reviewer #1: Cross-cultural adaptation and psychometric evaluation of the Yoruba version of

Oswestry disability index

Thank you for the opportunity to review the manuscript. I reviewed this manuscript carefully with a great interest. I respectfully provided my comments below

Methods

1: In methods section no description for validation( face, content, construct (EFA and CFA) and convergent) are provided. Please give more details about that.

2: Where did the 280150 samples for EFA and CFA come from? Did the authors do the power analysis for the sample sizes?

3: The choice of PCA to evaluate the factor structure of the ODI-Y appears somewhat rudimentary given the increasingly sophisticated statistically methods that have been used to evaluate the scale since the publication of the original article. It would be good to know why the author has chosen to use PCA rather than using structural equation models which have become much more commonly used to test the psychometric properties of scales.

4: What rotation model of EFA was used?

5: .Say more about the sample of 86 who completed the questionnaire twice. Who were they?

Data Analysis

1:Please report first the result validity next reliability

2 The values for RMSEA is higher and not adequate. Its cut off value is 0.08 according to Bentler & Hu

3: GFI= 0.87; AGFI=0.85: These values should at least be higher than 0.90. The model has inadequate model of fit

4: Lastly, factors should be retained using Parallel Analysis. Scree plot and Eigen values are traditional methods

The discussion part of this paper is not good. It would be better if you could write this part focusing on the results from the study

Reviewer #2: It seems that the translation and adaptation steps as well as the validity and reliability of the questionnaire have been done to an acceptable level. However, there are problems that should be addressed by the authors:

- In the first principal component analysis, three factors with a variance accounting for 62% were obtained. What was the authors' justification for the second principal component analysis with two factors and less variance coverage than the previous one (about 51%)? (lines: 247-283]

- Principal component analysis for construct validity was not performed in a single-factor model but was performed in confirmatory factor analysis; What is the reason for this.

- The amount reported to cover the variance of the second principal component analysis in the abstract (51.7%) does not match with main text (51.47%).

- Report the factor loading of all items in the exploratory factor analysis output table

- The population under study in exploratory and confirmatory factor analysis should be distinct. However, none of the model versions had the optimal fit indices in factor analysis. Therefore, it may be advisable not to perform a factor analysis in this article and perform it in a distinct population.

- The authors in the results section in the abstract have stated that: " the ODI-Y yielded a two-factor structure which accounted for 51.7% of the 54 variance but showed poor fit."

But in the discussion section they stated that: "the ODI was adapted into the Yoruba language and proved to have a 58 good factor structure and psychometric properties that replicated the results of other 59 obtainable versions." Don't these two terms contradict each other?

6. PLOS authors have the option to publish the peer review history of their article (what does this mean?). If published, this will include your full peer review and any attached files.

Reviewer #1: Yes: Razieh Bandari

Reviewer #2: No

---

## [Author Response · Author response to Decision Letter 0]

17 Dec 2019

Journal Requirements as requested by academic editor:

A: We have addressed this to the best of our ability. Thank you very much.

A: We included captions for our supporting information and updated all in-text citations to match.

3. Please carefully proofread your manuscript for typographical errors. For example, on page 8 “However, no less 50 participants be considered adequate…” should be “However, no less than 50 participants is considered adequate…”

A: We have proofread our manuscript and corrected all the typographical errors. Thank you.

4. Please state in your methods section the participant recruitment date

.A: The participant recruitment date (10/12/2015 and 10/01/2016) has been included in the methods section) (Page 5, line 109)

5. Please state the participants’ age range (in addition to their mean age).

A: The age range of the participants (35-70 years) has been included (Page 11, line 250).

6. Please amend your current ethics statement and methods section to address the following concerns: a) Did participants provide their written or verbal informed consent to participate in this study? b) If consent was verbal, please explain: i) Why was written consent not obtained? ii) How did you record/document participant consent? iii) Did the ethics committees/IRBs approve this consent procedure?

A: Thank you very much for highlighting this omission in our manuscript. The participants provided their written informed consent. The consent procedure was approved by the ethics committee/IRB (Page 5, line 110).

7. Please amend the manuscript submission data (via Edit Submission) to include author Ajibola Babatunde Oladiran.

A: This has been done. Thank you very much for pointing this out.

Reviewers' comments:

Reviewer's Responses to Questions

Comments to the Author

1. Is the manuscript technically sound, and do the data support the conclusions?

Reviewer #1: Partly

Reviewer #2: Partly

2. Has the statistical analysis been performed appropriately and rigorously?

Reviewer #1: Yes

Reviewer #2: No

3. Have the authors made all data underlying the findings in their manuscript fully available?

Reviewer #1: Yes

Reviewer #2: Yes

4. Is the manuscript presented in an intelligible fashion and written in standard English?

Reviewer #1: Yes

Reviewer #2: Yes

5. Review Comments to the Author

Reviewer #1: Cross-cultural adaptation and psychometric evaluation of the Yoruba version of Oswestry disability index

Thank you for the opportunity to review the manuscript. I reviewed this manuscript carefully with a great interest. I respectfully provided my comments below

Methods

1: In methods section no description for validation (face, content, construct (EFA and CFA) and convergent) are provided. Please give more details about that.

A: Thank you very much for pointing our attention to this; we have included more details about the EFA and convergent validity. We have had to remove the CFA following the suggestion of the second reviewer (Page 9, Line 205- page 10 Line 224).

2: Where did the 280150 samples for EFA and CFA come from? Did the authors do the power analysis for the sample sizes?

A: We didn’t do a power analysis. However, we adopted a minimum sample size for the EFA.

3: The choice of PCA to evaluate the factor structure of the ODI-Y appears somewhat rudimentary given the increasingly sophisticated statistically methods that have been used to evaluate the scale since the publication of the original article. It would be good to know why the author has chosen to use PCA rather than using structural equation models which have become much more commonly used to test the psychometric properties of scales.

A: Thank you very much for enlightening us about the superiority of structural equation modelling over PCA. Our sample was less than the 200 minimum sample size for the CFA and SEM. We had to use the PCA which our sample was adequate for. We hope to explore this suggestion subsequently.

 4: What rotation model of EFA was used?

A: We have included this in the manuscript (Promax rotation): Page 10, line 220)

5: .Say more about the sample of 86 who completed the questionnaire twice. Who were they?

A: We have included the mean age and gender of participants who took part in the retest phase “(55.9±11.0 years; 54.7% female)” (Page 9, line 195).

Data Analysis

1:Please report first the result validity next reliability

A: This had been done, thank you ( Page 12, lines 260-267)

2 The values for RMSEA is higher and not adequate. Its cut off value is 0.08 according to Bentler & Hu

3: GFI= 0.87; AGFI=0.85: These values should at least be higher than 0.90. The model has inadequate model of fit

A: We have removed the CFA analysis from the study due to the inadequacy of our sample size and the suggestion of the second reviewer.

4: Lastly, factors should be retained using Parallel Analysis. Scree plot and Eigen values are traditional methods

A: This had been done thank you very much for the great inputs to our manuscript (Page 10, line 219).

 The discussion part of this paper is not good. It would be better if you could write this part focusing on the results from the study

A: We have overhauled the discussion section as recommended. Thank you.

Reviewer #2: It seems that the translation and adaptation steps as well as the validity and reliability of the questionnaire have been done to an acceptable level. However, there are problems that should be addressed by the authors:

- In the first principal component analysis, three factors with a variance accounting for 62% were obtained. What was the authors' justification for the second principal component analysis with two factors and less variance coverage than the previous one (about 51%)? (lines: 247-283]

A: thank you very much for this important observation. We wanted a structure that could be explainable looking at the items loading in them. However we have decided to stick to the initial PCA analysis as it has more variance coverage and the factors can also be explained.

- Principal component analysis for construct validity was not performed in a single-factor model but was performed in confirmatory factor analysis; What is the reason for this.

A: Thank you very much. Events have overridden this as we have removed the CFA in its entirety from the study. 

- The amount reported to cover the variance of the second principal component analysis in the abstract (51.7%) does not match with main text (51.47%).

A: Thank you very much for noting this error. We have however expunged the second rotation and stuck to the first. 

- Report the factor loading of all items in the exploratory factor analysis output table

A: We have reported the factor loadings of all items in the exploratory factor analysis output table (Page 14, 291).

- The population under study in exploratory and confirmatory factor analysis should be distinct. However, none of the model versions had the optimal fit indices in factor analysis. Therefore, it may be advisable not to perform a factor analysis in this article and perform it in a distinct population.

A: thank you very much for the advice, this partly informed our removal of the CFA from our analysis

- The authors in the results section in the abstract have stated that: " the ODI-Y yielded a two-factor structure which accounted for 51.7% of the 54 variance but showed poor fit."

But in the discussion section they stated that: "the ODI was adapted into the Yoruba language and proved to have a 58 good factor structure and psychometric properties that replicated the results of other 59 obtainable versions." Don't these two terms contradict each other?

 A: Thank you very much. This was an error and we have adjusted the write up only to highlight the psychometric properties what were explored in the study.

---

## [Editor Report · Decision Letter 1]

26 Dec 2019

Cross-cultural adaptation and psychometric evaluation of the Yoruba version of Oswestry disability index

PONE-D-19-21244R1

Dear Dr. Idowu,

We are pleased to inform you that your manuscript has been judged scientifically suitable for publication and will be formally accepted for publication once it complies with all outstanding technical requirements.

With kind regards,

Ali Montazeri

Academic Editor

PLOS ONE
---

## [Editor Report · Acceptance letter]

30 Dec 2019

PONE-D-19-21244R1 

Cross-cultural adaptation and psychometric evaluation of the Yoruba version of Oswestry disability index 

Dear Dr. Idowu:

I am pleased to inform you that your manuscript has been deemed suitable for publication in PLOS ONE. Congratulations! Your manuscript is now with our production department. 

With kind regards,

on behalf of

Professor Ali Montazeri 

Academic Editor

PLOS ONE